# Exploring curvature noise in large-batch stochastic optimization

## Abstract

Using stochastic gradient descent (SGD) with large batch-sizes to train deep neural networks is an increasingly popular technique. By doing so, one can improve parallelization by scaling to multiple workers (GPUs) and hence leading to significant reductions in training time. Unfortunately, a major drawback is the so-called "generalization gap": large-batch training typically leads to a degradation in generalization performance of the model as compared to small-batch training. In this paper, we propose to correct this generalization gap by adding diagonal Fisher curvature noise to large-batch gradient updates. We provide a theoretical analysis of our method in the convex quadratic setting. Our empirical study with state-of-the-art deep learning models shows that our method not only improves the generalization performance in large-batch training, but furthermore, does so in a way where the training convergence remains desirable and the training duration is not elongated. We additionally connect our method to recent works on loss surface landscape in the experimental section.

## 1 Introduction

Modern datasets and network architectures in deep learning are becoming increasingly larger which results in longer training time. An ongoing challenge for both researchers and practitioners is how to scale up deep learning while keeping training time manageable. Using stochastic gradient descent (SGD) with large batch-sizes offers a potential avenue to address scalability issues. Increasing the batch-size used during training improves data parallelization (Goyal et al., 2017; You et al., 2018); it ensures multiple processors (GPUs) have sufficiently useful workload at each iteration hence reducing communication-to-cost ratio.

Unfortunately, a severe limitation to employing large-batch training in practice is the so-called "generalization gap". While large-batch yields considerable advantages over small-batch on training loss and error per parameter update, it has been verified empirically in LeCun et al. (2012); Keskar et al. (2016) that we have the opposite effect for test loss and error. To fully realize the benefits of using large-batches in the distributed synchronous setup, it is necessary to engineer large-batch training in a way such that this generalization gap can be closed without sacrificing too much the training performance. This is precisely the central objective of our paper.

In this paper, we propose to add diagonal Fisher curvature noise to large-batch gradient updates. We discuss the motivations behind our approach. Under the typical log-likelihood loss assumption, the difference of large-batch and small-batch gradients can be modeled as a Fisher noise. We can expect that adding this noise directly to large-batch gradients gives small-batch performance. While this remedies the generalization issues associated with large-batch training, the resulting convergence performance will be undesirable. To attain our end goal of designing a method which enjoys both desirable convergence and generalization performance, we reduce the amplitude of the noise by changing the covariance structure from full Fisher to diagonal Fisher. We find that in practice, this improves convergence considerably while maintaining good generalization.

We give a theoretical analysis of our proposed method in Section 3. This is done over the convex quadratic setting which often serves as an excellent "proxy" for neural network optimization (Martens, 2010). We additionally provide numerical experiments on standard deep-learning tasks in Section 4 to demonstrate the efficacy of our proposed method. There are two primary takeaways from our empirical analysis. First, we show adding diagonal Fisher noise to large-batch

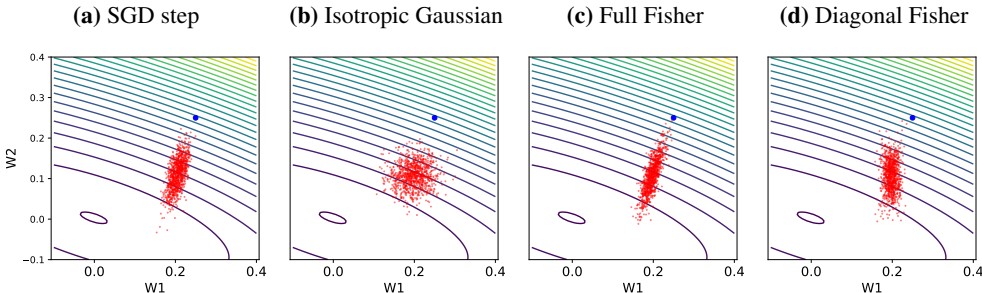

**Figure 1:** Noise structure in a simple two-dimensional regression problem. **a)** One-step SGD update. **b)** One-step SGD update with isotropic Gaussian ($\sigma = 0.1$) noise. **c)** One-step SGD update with full Fisher noise. **d)** One-step SGD update with diagonal Fisher noise. The full Fisher noise almost recovers the SGD noise. Observe that the full Fisher noise direction is perpendicular to the contours of the loss surface. Moreover, full Fisher exhibits slower convergence than diagonal Fisher; we refer to Section 3.2 for a more detailed analysis.

gradients does not hinder much the training performance; while at the same time giving a validation accuracy comparable to small-batch training, thereby correcting the "generalization gap". Second, this validation accuracy was attained in the same number of epochs used for small-batch training. This indicates that our method is data-efficient and does not require any lengthening of the optimization process.

## 2 PRELIMINARIES

### 2.1 BACKGROUND

**Excess risk decomposition.** Let $\mathcal{S} = \{(x_1, y_1), \ldots, (x_N, y_N)\}$ be a training set of $N$ input-target samples drawn i.i.d. from an unknown joint probability distribution $\mathcal{D}$. The family of classifiers of interest to us are neural network outputs $f(x_i, \theta)$, where $\theta \in \Theta \subset \mathbb{R}^d$ are parameters of the network and $\Theta$ here denotes the parameter space of the network. Let $\ell(f(x_i, \theta), y_i)$ be the loss function measuring the disagreement between outputs $f(x_i, \theta)$ and targets $y_i$. For convenience, we use the notation $\ell_i(\theta_k)$ to denote $\ell(f(x_i, \theta), y_i)$. The expected risk and empirical risk functions are defined to be

$$\mathscr{L}(\theta) \equiv \mathbb{E}_{(x,y) \sim \mathcal{D}}[\ell(f(x, \theta), y)], \ \mathcal{L}(\theta) \equiv \frac{1}{N} \sum_{i=1}^{N} \ell_i(\theta).$$

The standard technique to analyze the interplay between optimization and generalization in statistical learning theory is through excess risk decomposition. After $k$ iterations, the excess risk is defined as

$$\hat{\mathcal{L}}(\theta_k) \equiv \mathscr{L}(\theta_k) - \inf_{\theta \in \Theta} \mathscr{L}(\theta).$$

From Bottou & Bousquet (2008); Chen et al. (2018), the expected excess risk is upper-bounded by

$$\mathbb{E}_{\mathcal{S}}[\hat{\mathcal{L}}(\theta_k)] \leq \underbrace{\mathbb{E}_{\mathcal{S}}[\mathscr{L}(\theta_k) - \mathcal{L}(\theta_k)]}_{\mathcal{E}_{\text{gen}}} + \underbrace{\mathbb{E}_{\mathcal{S}}[\mathcal{L}(\theta_k) - \mathcal{L}(\theta^*)]}_{\mathcal{E}_{\text{opt}}}, \tag{1}$$

where $\theta^* = \arg\min_\theta \mathcal{L}(\theta)$ here is the empirical risk minimizer. The terms $\mathcal{E}_{\text{gen}}$ and $\mathcal{E}_{\text{opt}}$ are the expected generalization and optimization errors respectively. In the machine learning literature, optimization algorithms are often studied from one perspective: either optimization or generalization. The excess risk decomposition suggests that both aspects should be analyzed together (Chen et al., 2018; Bottou & Bousquet, 2008) since the goal of a good optimization-generalization algorithm in machine learning is to minimize the excess risk in the least amount of iterations.

**Related Work.** In the context of large-scale learning, stochastic algorithms are very popular compared to full-batch methods due to lower computational overhead (Bottou et al., 2018; Bottou, 1991). Despite a decay in optimization performance (slower rate of convergence in the convex case (Moulines & Bach, 2011; Bottou et al., 2018)), stochastic algorithms possess good generalization properties due to the inherent noise of their gradients. There are vast bodies of research literature devoted to understanding the connection between inherent noise and generalization; for example, scaling the learning rate or batch-size (Smith & Le, 2017; Goyal et al., 2017; Hoffer et al.,

2017) or studying the covariance structure of mini-batch gradients (Jastrzebski et al., 2017; Xing et al., 2018; Zhu et al., 2018; Li et al., 2015). In the non-convex setting, the inherent noise allows SGD to efficiently escape from saddle points or shallow local minima which tends to give poor generalization (Zhu et al., 2018; Daneshmand et al., 2018; Jin et al., 2017). The trade-off between optimization and generalization was also observed for large-batch versus small-batch training in deep learning: both large-batch and small-batch training can reach similar training loss after sufficient number of iterations. Since large-batch gradients have smaller variance, large-batch typically requires less iterations to reach an optimum. However, large-batch training usually has worse test performance compared to small-batch (Hoffer et al., 2017; Masters & Luschi, 2018; Keskar et al., 2016). It was recently observed that generalization and optimization should not be decoupled in deep learning (Neyshabur et al., 2014; 2017a;b). The choice of the batch-size has a direct impact on the trade-off between optimization and generalization.

In this paper, we aim to design a novel algorithm for large-batch training by increasing the variance of its gradients such that the resulting generalization performance matches small-batch gradient descent. Moreover, we require that this generalization performance be achieved within a number of iterations comparable to original large-batch.

## 2.2 MOTIVATION

We tackle the large-batch generalization problem through gradient noise injection. Let $B_L$ denote large-batch and $M_L = |B_L|$ denote size of the large-batch. Consider the following modification of large-batch SGD updates

$$\theta_{k+1} = \theta_k - \alpha_k \nabla \mathcal{L}_{M_L}(\theta_k) + \alpha_k D(\theta_k)\xi_{k+1}, \ \xi_{k+1} \sim \mathcal{N}(0, I_d). \tag{2}$$

where $\alpha_k$ is the learning rate, $\nabla \mathcal{L}_{M_L}(\theta_k) = \frac{1}{M_L}\sum_{i=1}^{M_L} \ell_i(\theta_k)$ is the large-batch gradient and $\mathcal{N}(0, I_d)$ is the multivariate Gaussian distribution with mean zero and identity convariance. We can interpret this modification as injecting a Gaussian noise with mean zero and covariance $D(\theta_k)D(\theta_k)^\top$ to the gradients. We now focus our attention on finding a suitable matrix $D(\theta_k)$ such that the algorithm in Eqn. 2 has desirable convergence and generalization performance.

**Intrinsic SGD noise.** Let $B \subset \mathcal{S}$ be a mini-batch drawn uniformly and without replacement from $\mathcal{S}$ and $M = |B|$ be the size of this chosen mini-batch. We can write the SGD update rule here as

$$\theta_{k+1} = \theta_k - \alpha_k \nabla \mathcal{L}_M(\theta_k)$$
$$= \theta_k - \alpha_k \nabla \mathcal{L}(\theta_k) + \alpha_k (\underbrace{\nabla \mathcal{L}(\theta_k) - \nabla \mathcal{L}_M(\theta_k)}_{\delta_k})$$

where $\nabla \mathcal{L}(\theta_k) = \frac{1}{N}\sum_{i=1}^{N} \nabla \ell_i(\theta_k)$ is the full-batch gradient. The difference $\delta_k = \nabla \mathcal{L}(\theta_k) - \nabla \mathcal{L}_M(\theta_k)$ is the intrinsic noise stemming from mini-batch gradients. The covariance of $\delta_k$ is

$$\mathrm{Cov}(\delta_k, \delta_k) = \left(\frac{N-M}{NM}\right) \cdot \frac{1}{N}\sum_{i=1}^{N}(\nabla \ell_i(\theta_k) - \nabla \mathcal{L}(\theta_k))(\nabla \ell_i(\theta_k) - \nabla \mathcal{L}(\theta_k))^\top, \tag{3}$$

This result can be found in Hu et al. (2017); Hoffer et al. (2017). Suppose that the loss function here is the negative log-likelihood, $\ell_i(\theta_k) = -\log p(y_i|x_i, \theta_k)$, where $p(y|x, \theta)$ is the density function for the model's predictive distribution.

**Fisher information matrix.** Given an input-target pair $(x, y)$, the Fisher information matrix is defined as the expectation of the outer product of log-likelihood gradients,

$$\mathbb{E}_{P_x, P_{y|x}}[\nabla \log p(y|x, \theta)\nabla \log p(y|x, \theta)^\top]. \tag{4}$$

The expectation here is taken with respect to the data distribution $P_x$ for inputs $x$ and the model's predictive distribution $P_{y|x}$ for targets $y$. The Fisher matrix can be realized as the second-order Taylor expansion of the KL-divergence on the space of predictive distributions, hence defining a Riemannian metric on this space (Amari, 1998; Martens, 2014). The Fisher matrix for neural networks captures the local curvature information of the parameter space. We give a detailed description of Fisher matrices for feed-forward and convolutional network architectures in Appendix C.

In practical situations, we often sample $y$ from the empirical training distribution rather than the predictive distribution. Therefore, we have the empirical Fisher (Martens, 2014)

$$\frac{1}{N}\sum_{(x_i, y_i)\in\mathcal{S}} \nabla \log p(y_i|x_i, \theta)\nabla \log p(y_i|x_i, \theta)^\top.$$

For the rest of this paper, unless otherwise specified, all mentions of "Fisher matrix" or $F(\theta)$ refers to the empirical Fisher.

**Naive method to close "generalization gap".** Having modeled the covariance of the mini-batch SGD noise as a Fisher matrix, we now consider a naive approach to close "generalization gap" in large-batch training. Let $B_S$ denote small-batch and $M_S = |B_S|$ denote the size of small-batch. Let us choose $D(\theta_k)$ in Eqn. 2 to be

$$D(\theta_k) = \sqrt{\frac{M_L - M_S}{M_L M_S}} \sqrt{F(\theta_k)}, \tag{5}$$

where $\sqrt{F(\theta_k)}$ is the square-root of Fisher, $\sqrt{F(\theta_k)}\sqrt{F(\theta_k)}^\top = F(\theta_k)$. The motivation here is that the difference between large-batch and small-batch gradients can be approximated as a Gaussian noise with mean zero and covariance $\frac{M_L - M_S}{M_L M_S} F(\theta_k)$ (Jastrzebski et al., 2017; Xing et al., 2018; Zhu et al., 2018; Li et al., 2015). However, there is an immediate issue with this naive approach: if the Gaussian approximation is reasonable, then this algorithm should have similar behavior as small-batch training which implies poor convergence performance. Indeed, as shown in the 2D convex example in Fig. 1c, adding full Fisher noise recovers SGD behavior. Furthermore, on the CIFAR-10 image classification task trained using ResNet44, we see in Fig. 3c that the training performance is almost identical to small-batch during the entire optimization process. Thus, this choice of $D(\theta_k)$ does not satisfy our requirement of maintaining desirable convergence.

## 3 METHOD

### 3.1 PROPOSED ALGORITHM

**Using diagonal Fisher.** We consider replacing $\sqrt{F(\theta_k)}$ with $\sqrt{\mathrm{diag}\,(F(\theta_k))}$ in Eqn. 5 for $D(\theta_k)$. Our proposed algorithm to correct the "generalization gap" in large-batch training is to inject diagonal Fisher noise to large-batch gradients,

$$\theta_{k+1} = \theta_k - \alpha_k \nabla \mathcal{L}_{M_L}(\theta_k) + \alpha_k \sqrt{\frac{M_L - M_S}{M_L M_S}} \sqrt{\mathrm{diag}\,(F(\theta_k))} \xi_{k+1}, \ \xi_{k+1} \sim \mathcal{N}(0, I_d).$$

The formal statement is given in Algorithm 1. Most of the empirical analysis of Algorithm 1 in Section 4 later uses feed-forward and convolutional networks. For completeness, we provide explicit expressions of diagonal Fisher for these architectures in Appendix C.

Changing the covariance structure in this way has important implications with regards to convergence and generalization performance. In the next subsection, we analyze our algorithm in a simple quadratic setting and compare it to the case where the covariance is given by the Fisher matrix. Even in this simple case, it is not obvious to compare the generalization performance between the two algorithms. In Appendix B, we provide an extensive theoretical discussion regarding how the choice of covariance structure influences generalization performance of Eqn. 2.

Working with the assumption that the generalization error is comparable between diagonal Fisher and full Fisher, the excess risk in Eqn. 1 can be minimized by focusing only on the optimization error. We prove in Theorem 3.1 that the discrepancy in convergence behavior can be measured by the difference of their respective Frobenius norms. In Fig. 2, we illustrate this behavior on a 2D toy problem where we compare the training trajectory of adding full Fisher noise versus diagonal full Fisher noise to the true gradient. As shown in the figure, diagonal Fisher converges faster than full Fisher with respect to the same number of iterations. In our experiments in Section 4, we observe that this phenomena carries over to the deep learning setting.

### 3.2 CASE-STUDY: CONVEX QUADRATIC EXAMPLE

In this subsection, we restrict our setting and take the loss function $\mathcal{L}(\theta)$ to be the convex quadratic,

$$\mathcal{L}(\theta) \equiv \frac{1}{2}\theta^\top A\theta,$$

where $A$ is a symmetric and positive-definite matrix. Observe that in this setting the Fisher and Hessian coincide, which is simply the matrix $A$.

---

**Algorithm 1** Adding diagonal Fisher noise to large-batch SGD. Differences from standard SGD are highlighted in blue

---

**Require:** Number of iterations $K$, initial step-size $\alpha_0$, large-batch $B_L$ of size $M_L$, small-batch $B_S$ of size $M_S$, initial condition $\theta_0 \in \Theta \subset \mathbb{R}^d$

    **for** $k = 1$ to $K$ **do**
        $Z_k \sim \mathcal{N}(0, I_d)$
        $\epsilon_k = \alpha_k \sqrt{\frac{M_L - M_S}{M_L M_S}} \sqrt{\text{diag}\left(F(\theta_k)\right)} Z_k$
        $\theta_{k+1} = \theta_k - \alpha_k \nabla \mathcal{L}_{M_L}(\theta_k) + \epsilon_k$
    **end for**

---

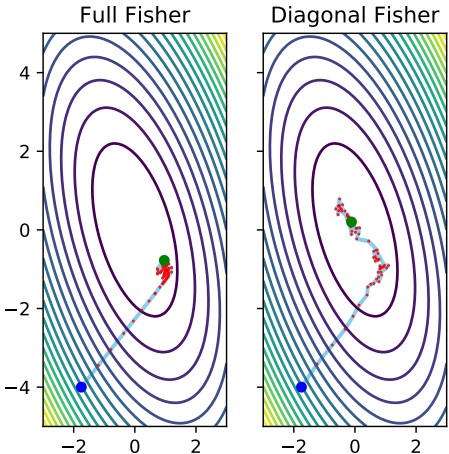

We stress here that approximating the loss surface of a neural network with a quadratic often serves as a fertile "testing ground" when introducing new methodologies in deep learning. Analyzing the toy quadratic problem has led to important advances; for example, in learning rate scheduling (Schaul et al., 2013) and formulating SGD as approximate Bayesian inference (Mandt et al., 2017). With regards to optimization, Martens (2010) observed that much of the difficulty in neural network optimization can be captured using quadratic models.

**Figure 2:** Trajectory using full Fisher noise versus diagonal Fisher noise for the algorithm in Eqn. 2 used to minimize a two-dimensional quadratic function. Blue dot indicates the initial parameter value and the green dot shows the final parameter value. We used a learning rate of 0.1 for 500 iterations (plotting every 10 iterations). Observe that adding diagonal Fisher to the true gradient achieves faster convergence than full Fisher.

We focus now on the algorithm in Eqn. 2 and choose a constant $d \times d$ covariance matrix $C$. The following theorem, adapted from Bottou et al. (2018), analyzes the convergence of this optimization method. The proof is relegated to Appendix A.

**Theorem 3.1.** *Let $\lambda_{\max}$ and $\lambda_{\min}$ denote the maximum and minimum eigenvalue of $A$ respectively. For a chosen $\alpha_0 \leq \lambda_{\max}^{-1}$, suppose that we run the algorithm in Eqn. 2 according to the decaying step-size sequence*

$$\alpha_k = \frac{2}{(k + \gamma)\lambda_{\min}},$$

*for all $k \in \mathbb{N}_{>0}$ and where $\gamma$ is chosen such that $\alpha_k \leq \alpha_0$. Then for all $k \in \mathbb{N}$,*

$$\mathbb{E}[\mathcal{L}(\theta_k)] \leq \frac{\nu}{k + \gamma} \quad \text{where } \nu = \max\left(\frac{2\,\text{Tr}(C^\top A C)}{\lambda_{\min}^2}, \gamma \mathcal{L}(\theta_0)\right).$$

We make a few observations regarding this bound. First, the convergence rate is optimal when $C = 0$. However, in this case, there is no noise and so we obtain no regularization benefits which leads to poor generalization. A more formal discussion is given at the end of Appendix B where if we use a scaling factor $C_\lambda := \lambda C$; as $\lambda \to 0$, the expected generalization error becomes worse.

The second observation to note is that the term of importance in this theorem is $\text{Tr}(C^\top A C)$. While the overall convergence rate of the algorithm is $O(1/k)$, the discrepancy in convergence performance for different choices of the matrix $C$ rests entirely on this term. We analyze two specific cases which are relevant for us: the first case where $C$ is square-root of $A$, $C = \sqrt{A}$, and the second case where $C$ is the square-root of the diagonal of $A$, $C = \sqrt{\text{diag}(A)}$. For the first, we have

$$\text{Tr}(C^\top A C) = \text{Tr}(A^2) = \|A\|_{\text{Frob}}^2.$$

For the latter case, we have

$$\text{Tr}(C^\top A C) = \text{Tr}(\text{diag}(A)^2) = \|\text{diag}(A)\|_{\text{Frob}}^2.$$

Thus, the difference in training performance between the two cases can be measured by the difference of their respective Frobenius norms.

### 3.3 SAMPLING RANDOM VECTOR WITH FISHER COVARIANCE

We describe a method to sample a random vector with mean zero and approximately Fisher covariance to avoid computing directly the square root of the empirical Fisher. Let $M$ be the size of the mini-batch and from the $M$-forward passes, we obtain prediction $f(x, \theta)$ and then from the backward passes, we obtain the back-propagated gradients $\nabla \ell_1, \ldots, \nabla \ell_M$ for each data-point. Consider independent random variables $\sigma_1, \ldots, \sigma_M$ drawn from Rademacher distribution, i.e., $\mathbb{P}(\sigma_i = 1) = \mathbb{P}(\sigma_i = -1) = 1/2$. Then, the mean $\mathbb{E}_\sigma[\sum_{i=1}^M \sigma_i \nabla \ell_i] = 0$. The covariance is empirical Fisher.

## 4 EXPERIMENTS

We first empirically compute the Frobenius norm of full Fisher and diagonal Fisher, followed by the marginal variance of the gradients for a variety of training regimes. We also compare the convergence speed of diagonal Fisher noise and full Fisher noise; the convergence speed here is measured with respect to the number of parameter updates. In Section 4.2, we compute the maximum eigenlue of the Hessian with respect to the final model parameters of different training regimes. The purpose of this is to empirically relate our methodology with the curvature of the loss surface landscape. In Section 4.3, we give the generalization performance of each method discussed previously.

Throughout our experiments, large-batch size **LB** is set to 4096 and small-batch size **SB** is set to 128 by default. The network architectures we use are fully-connected networks, shallow convolutional networks (LeNet (LeCun et al., 1998), AlexNet (Krizhevsky et al., 2012)), and deep convolutional networks (VGG16 (Simonyan & Zisserman, 2014), ResNet44 (He et al., 2016)). These models are evaluated on the standard deep-learning datasets: MNIST, Fashion-MNIST (LeCun et al., 1998; Xiao et al., 2017), CIFAR-10 and CIFAR-100 (Krizhevsky & Hinton, 2009).

### 4.1 CONVERGENCE EXPERIMENTS

In the convex quadratic setting in Section 3.2, we observed that full-batch gradient descent with diagonal Fisher noise enjoyed faster convergence than full Fisher noise. This was characterized by the difference of their Frobenius norms. We now give an empirical verification of this phenomena in the non-convex setting of deep neural networks. We compute the Frobenius norms during the training of a ResNet44 network on CIFAR-10. Fig. 3a shows that the full Fisher matrix has much larger Frobenius norm than the diagonal Fisher matrix, which suggests that using diagonal Fisher noise should have faster convergence than full Fisher noise in the deep neural network setting. To justify this intuition, we analyze the training loss (per parameter update) of ResNet44 (CIFAR-10) on the following four regimes: **LB**, **SB**, **LB** with diagonal Fisher noise and **LB** with full Fisher noise. For a fair comparison, we use the same learning rate schedule for all four regimes. Fig. 3c shows that **LB** with diagonal Fisher noise trains much faster than **LB** with full Fisher. More interestingly, **LB** with full Fisher matches the convergence performance of **SB**, indicating that the intrinsic noise of **SB** is accurately modeled by the Fisher. In contrast, **LB** with diagonal Fisher attains a convergence similar to **LB**, showing that our approach preserves the desired convergence behavior of **LB**.

Next, we give an estimation of the marginal variance of gradients for the four regimes mentioned above as well as **LB** with K-FAC noise. K-FAC (Martens & Grosse, 2015) is a block-diagonal approximation of the Fisher matrix used as a second-order optimization method in deep learning. The purpose of this experiment is to show despite the fact that **LB** with diagonal Fisher trains much faster than **LB** with full Fisher, they share roughly the same marginal variance of the gradients. This suggests that the off-diagonal elements of the Fisher matrix is the key reason for slow convergence. The experiment is performed as follows: we freeze a partially-trained network and compute Monte-Carlo estimates of gradient variance with respect to each parameter over different mini-batches. This variance is then averaged over the parameters within each layer. The results are presented in Fig. 3b. We find that **LB** with diagonal Fisher noise and **LB** with full Fisher noise give roughly the same scale of marginal gradients variance.

### 4.2 MAXIMUM EIGENVALUE COMPARISON

While the relationship between loss surface curvature and generalization is not completely explicit, numerous works have suggested that the maximum eigenvalue of Hessian is possibly correlated

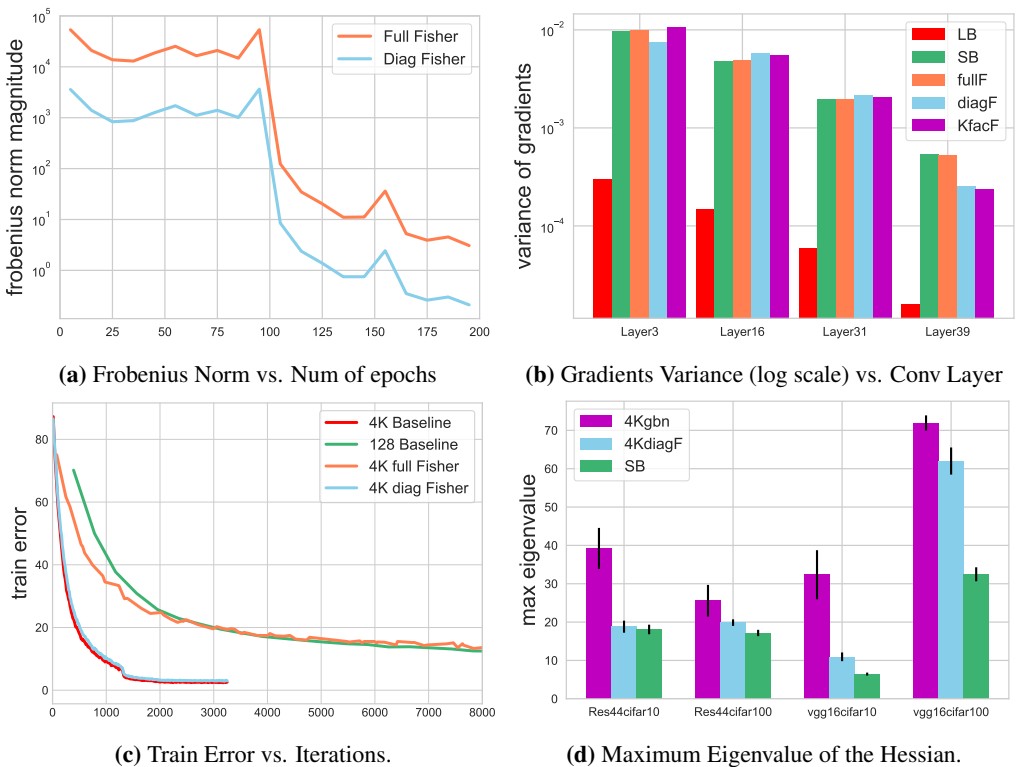

**(a)** Frobenius Norm vs. Num of epochs

**(b)** Gradients Variance (log scale) vs. Conv Layer

**(c)** Train Error vs. Iterations.

**(d)** Maximum Eigenvalue of the Hessian.

**Figure 3: a)** Frobenius norm of full Fisher matrix and diagonal Fisher matrix. The model is trained on ResNet44 with CIFAR-10. **b)** Estimation of gradient variances for randomly selected convolutional layers on ResNet44 (CIFAR-10). **fullF: LB** with full Fisher noise. **diagF: LB** with diagonal Fisher noise. **KfacF: LB** with K-FAC noise. **c)** Training error with respect to iterations between **SB**, **LB**, **LB** with full Fisher noise and diagonal Fisher noise on ResNet44 (CIFAR-10). All of the above are trained with the same learning rate. **d)** The maximum eigenvalue of the Hessian matrix at the end of training. **4Kgbn: LB** with Ghost-BN. **4KdiagF: LB** with diagonal Fisher noise.

with generalization performance (Keskar et al., 2016; Chaudhari et al., 2016; Chaudhari & Soatto, 2017; Yoshida & Miyato, 2017; Xing et al., 2018). To situate our method with this line of research, we compute the maximum eigenvalue of the Hessian of the final model for the following three regimes: **LB** with diagonal Fisher noise, **LB** with ghost batch-normalization, and **SB**. Ghost batch-normalization (GBN) is an adaptation of usual batch-normalization introduced in Hoffer et al. (2017) to close generalization gap. In Fig. 3b, we find that **LB** with diagonal Fisher gives smaller maximum eigenvalue than **LB** with GBN.

Computing maximum eigenvalue without any modification to the model gives inconsistent estimates even between different runs of the same training configuration. To make the maximum eigenvalue of the Hessian matrix comparable over different training trajectories, the Hessian needs to be invariant under typical weight reparameterizations such as affine transformations (Liao et al., 2018). To achieve this, we make the following modification to the trained model: (1) For the layer with batch normalization, we can just push the batch-norm layer parameters and running statistics into the layerwise parameter space so that the the layer is invariant under affine transformation; and (2) For the layer without batch normalization, reparameterization changes the prediction confidence while the prediction remains the same. So we train a temperature parameter on the cross-entropy loss on the test data set, this encourages the model to make a calibrated prediction (prevent it from being over-confident). In Fig. 3d, we give the error bar of the maximum eigenvalue of the Hessian over different runs, which indicates the modification gives a roughly consistent estimate.

### 4.3 RESULTS

In this subsection, we provide the validation accuracy for a number of training regimes. We point out that we do not experiment **LB** with full Fisher due to its exceedingly long training time. This can be seen from Fig. 3c where on ResNet44 (CIFAR-10), full Fisher does not achieve good convergence even after 8000 iterations. We use the baseline in Hoffer et al. (2017), where **LB** is trained using GBN with learning rate adaptation. However, instead of using square-root scaling of the learning rate in Hoffer et al. (2017), we use linear scaling in conjunction with a warmup scheme suggested by Goyal et al. (2017). We found that this improves the baseline reported in Hoffer et al. (2017).

We experimented with the following regimes: **LB** with GBN (our baseline), **LB** with GBN and trace Fisher noise (where $D(\theta_k)$ in Eqn. 2 is taken to be the square-root of the trace of Fisher $\sqrt{\mathrm{Tr}(F(\theta_k))}$), **LB** with GBN and diagonal Fisher noise, and **SB**. All regimes were trained for the same number of epochs. For the first three regimes involving **LB**, we terminate the noise at epoch 50 and use standard **LB** for the remainder of training. The reasoning behind this comes from the work of Smith et al. (2017) which suggests that using noise only matters in the early stages of the optimization procedure. We give further explanation in Appendix E.

The final validation accuracies of each method are reported in Table 1. While it is true that adding diagonal Fisher noise cannot completely close the "generalization gap" in some cases, we can see from Table 1 that doing so yields definite improvements. We point out that we explored other regimes such as injecting multiplicative Guassian noise with constant diagonal covariance (Hoffer et al., 2017), but found that they all perform no better than our baseline (**LB** with GBN). In addition, we experimented with K-FAC noise but this did not give additional benefits over diagonal Fisher noise. We defer to Appendix E for exact details of these supplemental experiments.

**Table 1:** Validation accuracy results on classification tasks. GBN stands for Ghost-BN. Fisher Trace+GBN stands for LB + Ghost-BN with isotropic Gaussian noise scaled by square-root of trace of Fisher. Diag-F+GBN stands for LB + Ghost-BN with diagonal Fisher noise. All methods in each row are trained with the same number of epochs. Confidence interval is computed over 3 random seeds. **LB** with full Fisher requires same number of updates as **SB**, which renders it impractical for most of the models for which we work with. While this is indeed infeasible to compute for all the models, we note that **LB** with full Fisher reaches roughly the same validation accuracy (93.22) as **SB** in the case of ResNet44 (CIFAR-10).

| Dataset | Network | SB | LB+GBN | Fisher Trace+GBN | Diag-F+GBN |
|---|---|---|---|---|---|
| MNIST | MLP | 98.10 | 97.94 | 98.08 | **98.08** |
| MNIST | LeNet | 99.10 | 98.85 | 99.02 | **99.10** |
| FASHION-MNIST | LeNet | 91.10 | 88.89 | 90.29 | **90.77** |
| CIFAR-10 | Alexnet | 87.80 | $86.41 \pm 0.18$ | N/A | $\mathbf{87.30 \pm 0.28}$ |
| CIFAR-100 | Alexnet | 59.21 | $56.75 \pm 0.18$ | N/A | $\mathbf{58.68 \pm 0.40}$ |
| CIFAR-10 | VGG16 | 93.25 | $91.78 \pm 0.29$ | $92.81 \pm 0.10$ | $\mathbf{93.15 \pm 0.05}$ |
| CIFAR-100 | VGG16 | 72.83 | $69.44 \pm 0.27$ | $71.26 \pm 0.09$ | $\mathbf{71.94 \pm 0.14}$ |
| CIFAR-10 | ResNet44 | 93.42 | $91.92 \pm 0.29$ | $92.31 \pm 0.02$ | $\mathbf{92.72 \pm 0.15}$ |
| CIFAR-100 | ResNet44x2 | 75.55 | $73.11 \pm 0.22$ | $73.62 \pm 0.15$ | $\mathbf{74.10 \pm 0.18}$ |

## 5 CONCLUSION

In this paper, we explored in depth the relationship between curvature noise and stochastic optimization. We proposed a method to engineer large-batch training such that we retain fast convergence performance while achieving significant gains in generalization. In addition, we highlighted the importance of noise covariance structure on optimization and generalization. An interesting future direction would be to further understand both empirically and theoretically how different noise covariance structures impact optimization and generalization performance of a learning algorithm.

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

## A  PROOF OF THEOREM 3.1

The proof of this theorem follows the spirit of Bottou et al. (2018). The algorithm

$$\theta_{k+1} = \theta_k - \alpha_k A\theta_k + \alpha_k C\xi_{k+1},\ \xi_{k+1} \sim \mathcal{N}(0, I_d).$$

falls into the Robbins-Monro setting where the true gradient is perturbed by random noise. This perturbation can be considered as a martingale difference in the sense that

$$\mathbb{E}[C\xi_{k+1}|\mathcal{F}_k] = 0$$

where $(\mathcal{F}_k)_{k\in\mathbb{N}}$ is a increasing filtration generated by the sequence of parameters $(\theta_k)_{k\in\mathbb{N}}$. When the step size is constant $\alpha_k = \alpha$ for all $k$, it corresponds to the Euler discretization of a gradient flow with random perturbation. We begin the proof by considering the equality,

$$\mathcal{L}(\theta_{k+1}) = \mathcal{L}(\theta_k) + \langle \mathcal{L}(\theta_k), \theta_{k+1} - \theta_k \rangle + \frac{1}{2}(\theta_{k+1} - \theta_k)^\top \nabla^2 \mathcal{L}(\theta_k)(\theta_{k+1} - \theta_k).$$

Using the fact that $\nabla\mathcal{L}(\theta_k) = A\theta_k$, $\nabla^2\mathcal{L}(\theta_k) = A$, and from the definition of $\theta_{k+1}$, we can rewrite the above equation as

$$\mathcal{L}(\theta_{k+1}) = \mathcal{L}(\theta_k) + \langle A\theta_k, -\alpha_k A\theta_k + \alpha_k C\xi_{k+1} \rangle + \frac{1}{2}\|\alpha_k A\theta_k - \alpha_k C\xi_{k+1}\|_A^2.$$

Now, taking the conditional expectation $\mathbb{E}[\cdot|\mathcal{F}_k]$ on both sides of the equality, we obtain by independence of the noise $\xi_{k+1}$ to $\mathcal{F}_k$

$$\mathbb{E}[\mathcal{L}(\theta_{k+1})|\mathcal{F}_k] = \mathcal{L}(\theta_k) - \alpha_k\|A\theta_k\|_2^2 + \frac{\alpha_k^2}{2}\|A\theta_k\|_A^2 + \frac{\alpha_k^2}{2}\mathbb{E}[\|C\xi_{k+1}\|_A^2] \tag{6}$$

A simple computation shows

$$\begin{aligned}
\mathbb{E}[\|C\xi_{k+1}\|_A^2] &= \mathbb{E}[(C\xi_{k+1})^\top A(C\xi_{k+1})] \\
&= \mathbb{E}[\xi_{k+1}^\top C^\top A C\xi_{k+1}] \\
&= \mathrm{Tr}(C^\top A C)
\end{aligned} \tag{7}$$

Moreover, we have

$$
\begin{aligned}
\|A\theta_k\|_A^2 &= (\theta_k^\top A)A(A\theta_k) \\
&= A\|A\theta_k\|_2^2 \\
&\leq \lambda_{\max}\|A\theta_k\|_2^2.
\end{aligned}
\tag{8}
$$

Using the results in Eqns. 7 and 8 as well as the assumption on the step-size schedule for all $k$: $\alpha_k < \alpha_0 < \frac{1}{\lambda_{\max}}$, we rewrite Eqn. 6 as

$$
\begin{aligned}
\mathbb{E}[\mathcal{L}(\theta_{k+1})|\mathcal{F}_k] &\leq \mathcal{L}(\theta_k) + \left(\frac{\alpha_k}{2}\lambda_{\max} - 1\right)\alpha_k\|A\theta_k\|_2^2 + \frac{\alpha_k^2}{2}\operatorname{Tr}(C^\top AC) \\
&\leq \mathcal{L}(\theta_k) - \frac{\alpha_k}{2}\|A\theta_k\|_2^2 + \frac{\alpha_k^2}{2}\operatorname{Tr}(C^\top AC).
\end{aligned}
\tag{9}
$$

Furthermore,

$$
\|A\theta_k\|_2^2 = A(\theta_k^\top A\theta_k) \geq \lambda_{\min}\|\theta_k\|_A^2 = 2\lambda_{\min}\mathcal{L}(\theta_k)
$$

Using this above fact and then taking the expectation of Eqn. 9 leads to

$$
\mathbb{E}[\mathcal{L}(\theta_{k+1})] \leq (1 - \alpha_k\lambda_{\min})\mathbb{E}[\mathcal{L}(\theta_k)] + \frac{\alpha_k^2}{2}\operatorname{Tr}(C^\top AC).
$$

We proceed by induction to prove the final result. By definition of $\nu$, the result is obvious for $k = 0$. For the inductive step, suppose that the induction hypothesis holds for $k$, i.e.,

$$
\alpha_k = \frac{2}{(k+\gamma)\lambda_{\min}}, \quad \mathbb{E}[\mathcal{L}(\theta_k)] \leq \frac{\nu}{k+\gamma}.
$$

We prove the $k + 1$ case.

$$
\begin{aligned}
\mathbb{E}[\mathcal{L}(\theta_{k+1})] &\leq \left(1 - \frac{2}{k+\gamma}\right)\frac{\nu}{k+\gamma} + \frac{2}{(k+\gamma)^2\lambda_{\min}^2}\operatorname{Tr}(C^\top AC) \\
&\leq \frac{\nu}{(k+\gamma+1)}
\end{aligned}
$$

This comes from the definition of $\nu$ and also the inequality $(k+\gamma-1)(k+\gamma+1) \leq (k+\gamma)^2$. This conclude the proof. $\qquad\square$

# B  RELATIONSHIP BETWEEN CHOICE OF COVARIANCE STRUCTURE AND GENERALIZATION

As in Section 3.2, we work entirely in the convex quadratic setting. Consider the algorithm

$$
\theta_{k+1} = \theta_k - \alpha_k\nabla\mathcal{L}(\theta_k) + \alpha_k C\xi_{k+1}, \ \xi_{k+1} \sim \mathcal{N}(0, I_d).
\tag{10}
$$

Our aim in this section is to provide some theoretical discussions on how the choice of covariance structure $C$ influences the generalization behavior of the above algorithm.

**Uniform stability.** Uniform stability (Bousquet & Elisseeff, 2002) is one of the most common techniques used in statistical learning theory to study generalization of a learning algorithm. Intuitively speaking, uniform stability measures how sensitive an algorithm is to perturbations of the sampling data. The more stable an algorithm is, the better its generalization will be. Recently, the uniform stability of several algorithms has been investigated for stochastic gradient methods (Hardt et al., 2015) or stochastic gradient Langevin dynamics algorithm (Mou et al., 2017; Raginsky et al., 2017). We present the precise definition.

**Definition B.1** (Uniform stability). A randomized algorithm $\mathcal{A}$ is $\epsilon$-stable if for all data sets $\mathcal{S}$ and $\mathcal{S}'$ where $\mathcal{S}$ and $\mathcal{S}'$ differ in at most one sample, we have

$$
\sup_{(x,y)} |\mathbb{E}_{\mathcal{A}}[\mathcal{L}(\theta_{\mathcal{S}}) - \mathcal{L}(\theta_{\mathcal{S}'})]| \leq \epsilon,
$$

where $\mathcal{L}(\theta_{\mathcal{S}})$ and $\mathcal{L}(\theta_{\mathcal{S}'})$ highlight the dependence of parameters on sampling datasets. The supremum is taken over input-target pairs $(x, y)$ belonging to the sample domain.

The following theorem from Bousquet & Elisseeff (2002) shows that uniform stability implies generalization.

**Theorem B.2** (Generalization in expectation). *Let $\mathcal{A}$ be a randomized algorithm which is $\epsilon$-uniformly stable, then*

$$|\mathbb{E}_{\mathcal{A}}[\mathcal{E}_{\text{gen}}]| \leq \epsilon,$$

*where $\mathcal{E}_{\text{gen}}$ is the expected generalization error as defined in Eqn. 1.*

**Continuous-time dynamics.** We like to use the uniform stability framework to analyze generalization properties of Eqn. 10. To do this, we borrow ideas from the recent work of Mou et al. (2017) which give uniform stability bounds for Stochastic Gradient Langevin Dynamics (SGLD) in non-convex learning. While the authors in that work give uniform stability bounds in both the discrete-time and continuous-time setting, we work with the continuous setting since this conveys relevant ideas while minimizing technical complications. The key takeaway from Mou et al. (2017) is that uniform stability of SGLD may be bounded in the following way

$$\epsilon_{\text{SGLD}} \leq \sup_{\mathcal{S}, \mathcal{S}'} \sqrt{H^2(\pi_t, \pi_t')}. \tag{11}$$

Here, $\pi_t$ and $\pi_t'$ are the distributions on parameters $\theta$ trained on the datasets $\mathcal{S}$ and $\mathcal{S}'$. The $H^2$ refers to the Hellinger distance.

We now proceed to mirror the approach of Mou et al. (2017) for Eqn. 10. Our usage of stochastic differential equations will be very soft but we refer to reader to Gardiner (2009); Pavliotis (2014) for necessary backgrounds. For the two datasets $\mathcal{S}$ and $\mathcal{S}'$, the continuous-time analogue of Eqn. 10 are Ornstein-Uhlenbeck processes (Uhlenbeck & Ornstein, 1930):

$$d\theta_{\mathcal{S}}(t) = -A_{\mathcal{S}}\theta_{\mathcal{S}}(t) + \sqrt{\alpha}C_{\mathcal{S}}dW(t)$$
$$d\theta_{\mathcal{S}'}(t) = -A_{\mathcal{S}'}\theta_{\mathcal{S}'}(t) + \sqrt{\alpha}C_{\mathcal{S}'}dW(t).$$

The solution is given by

$$\theta_{\mathcal{S}}(t) = e^{-A_{\mathcal{S}}t}\theta_{\mathcal{S}}(0) + \sqrt{\alpha}\int_0^t e^{-A_{\mathcal{S}}(t-u)}C_{\mathcal{S}}dW(u),$$

In fact, this yields the Gaussian distribution

$$\theta_{\mathcal{S}}(t) \sim \mathcal{N}(\mu_{\mathcal{S}}(t), \Sigma_{\mathcal{S}}(t)),$$

where

$$\mu_{\mathcal{S}}(t) = e^{-A_{\mathcal{S}}t}\theta_{\mathcal{S}}(0)$$

and $\Sigma_{\mathcal{S}}(t)$ satisfies the Ricatti equation,

$$\frac{d}{dt}\Sigma_{\mathcal{S}}(t) = -(A_{\mathcal{S}}\Sigma_{\mathcal{S}}(t) + \Sigma_{\mathcal{S}}(t)A_{\mathcal{S}}) + \alpha C_{\mathcal{S}}C_{\mathcal{S}}^{\top}.$$

Observe that $A_{\mathcal{S}}$ is symmetric and positive-definite which means that it admits a diagonalization $A_{\mathcal{S}} = P_{\mathcal{S}}D_{\mathcal{S}}P_{\mathcal{S}}^{-1}$. Solving the equation for the covariance matrix gives

$$\Sigma_{\mathcal{S}}(t) = \alpha P_{\mathcal{S}}\left(\int_0^t e^{-D_{\mathcal{S}}(t-u)}P_{\mathcal{S}}^{-1}C_{\mathcal{S}}C_{\mathcal{S}}^{\top}P_{\mathcal{S}}e^{-D_{\mathcal{S}}(t-u)}du\right)P_{\mathcal{S}}^{-1}. \tag{12}$$

We are in the position to directly apply the framework of (Mou et al., 2017). Choosing $\pi_t$ and $\pi_{t'}$ in Eqn. 11 to be the Gaussians $\mathcal{N}(\mu_{\mathcal{S}}(t), \Sigma_{\mathcal{S}}(t))$ and $\mathcal{N}(\mu_{\mathcal{S}'}(t), \Sigma_{\mathcal{S}'}(t))$ respectively, we obtain a uniform stability bound for Eqn. 10. We compute the right-hand side of the bound to get derive insights on generalization. Using the standard formula for Hellinger distance between two Gaussians, we have

$$H^2(\pi_t, \pi_t') = 1 - \frac{\det(\Sigma_{\mathcal{S}})^{\frac{1}{4}}\det(\Sigma_{\mathcal{S}'})^{\frac{1}{4}}}{\det(\frac{\Sigma_{\mathcal{S}}+\Sigma_{\mathcal{S}'}}{2})^{\frac{1}{2}}}\exp\left\{-\frac{1}{8}(\mu_{\mathcal{S}} - \mu_{\mathcal{S}'})^{\top}\left(\frac{\Sigma_{\mathcal{S}}+\Sigma_{\mathcal{S}'}}{2}\right)^{-1}(\mu_{\mathcal{S}} - \mu_{\mathcal{S}'})\right\} \tag{13}$$

**Choosing the noise covariance.** From Eqn. 13 above, it is evident that to ensure good generalization error for Eqn. 10, we want to choose a covariance $C_{\mathcal{S}}$ such that the Hellinger distance $H$

is minimized. Since we are working within the uniform stability framework, a good choice of $C_{\mathcal{S}}$ should be one where Eqn. 10 becomes less data-dependent. This is intuitive after all – the less data-dependent an algorithm is; the better suited it should be for generalization.

We study Eqn. 13. Note that as time $t \to \infty$, the exponential term goes to 1. Hence, we focus our attention on the ratio of the determinants. Suppose that we choose $C_{\mathcal{S}} = \sqrt{A_{\mathcal{S}}}$. Since $A_{\mathcal{S}}$ is the Fisher for this convex quadratic example, Eqn. 10 is essentially the naive method given in Section 2.2. Simplifying the determinant of $\Sigma_{\mathcal{S}}(t)$ in this case,

$$\det(\Sigma_{\mathcal{S}}(t)) = \left(\frac{\alpha}{2}\right)^d \det(I_d - e^{-2D_{\mathcal{S}}t})$$

Suppose that we choose $C = I_d$. Proceeding analogously,

$$\det(\Sigma_{\mathcal{S}}(t)) = \left(\frac{\alpha}{2}\right)^d \frac{\det(I_d - e^{-2D_{\mathcal{S}}t})}{\det(D_{\mathcal{S}})}$$

We can think of choosing $C = I_d$ or $C = \sqrt{A}$ to be extreme cases and it is interesting to observe that the Hellinger distance is more sensitive to dataset perturbation when $C = I_d$. Our proposed method of this paper was to choose $C = \sqrt{\operatorname{diag}(A)}$ and experiments in Section 4 seem to suggest that choosing the square-root of diagonal captures much of the generalization behavior of full Fisher. Understanding precisely why this is the case poses an interesting research direction to pursue in the future.

A simple scaling argument also highlights the importance of the trade-off between optimization and generalization. Consider $C_{\lambda} = \lambda C$. Then Theorem 3.1 suggests to take $\lambda$ small to reduce the variance and improve convergence. However, in that case $\Sigma_{\lambda} = \lambda^2 \Sigma$ where $\Sigma$ is given by the Eqn. 12 for $C$ and

$$H^2(\pi_t, \pi_t') = 1 - \frac{\det(\Sigma_{\mathcal{S}})^{\frac{1}{4}} \det(\Sigma_{\mathcal{S}'})^{\frac{1}{4}}}{\det(\frac{\Sigma_{\mathcal{S}} + \Sigma_{\mathcal{S}'}}{2})^{\frac{1}{2}}} \exp\left\{ -\frac{1}{8\lambda^2} (\mu_{\mathcal{S}} - \mu_{\mathcal{S}'})^\top \left(\frac{\Sigma_{\mathcal{S}} + \Sigma_{\mathcal{S}'}}{2}\right)^{-1} (\mu_{\mathcal{S}} - \mu_{\mathcal{S}'}) \right\}$$

and the Hellinger distance get close to one in the limit of small $\lambda$ (which intuitively corresponds to the large batch situation).

## C   Fisher information matrix for deep neural networks

In this section, we give a formal description of the true Fisher information matrix, rather than the empirical version, for both feed-forward networks and convolutional networks. In addition, we give the diagonal expression for both networks.

### C.1   Feed-forward networks

Consider a feed-forward network with $L$ layers. At each layer $i \in \{1, \ldots, L\}$, the network computation is given by

$$z_i = W_i a_{i-1}$$
$$a_i = \phi_i(z_i),$$

where $a_{i-1}$ is an activation vector, $z_i$ is a pre-activation vector, $W_i$ is the weight matrix, and $\phi_i : \mathbb{R} \to \mathbb{R}$ is a nonlinear activation function applied coordinate-wise. Let $w$ be the parameter vector of network obtained by vectorizing and then concatenating all the weight matrices $W_i$,

$$w = [\operatorname{vec}(W_1)^\top \ \operatorname{vec}(W_2)^\top \ \ldots \ \operatorname{vec}(W_L)^\top]^\top.$$

Furthermore, let $\mathcal{D}v = \nabla_v \log p(y|x, w)$ denote the log-likelihood gradient. Using backpropagation, we have a decomposition of the log-likelihood gradient $\mathcal{D}W_i$ into the outer product:

$$\mathcal{D}W_i = g_i a_{i-1}^\top,$$

where $g_i = \mathcal{D}z_i$ are pre-activation derivatives. The Fisher matrix $F(w)$ of this feed-forward network is a $L \times L$ matrix where each $(i, j)$ block is given by

$$F_{i,j}(w) = \mathbb{E}[\operatorname{vec}(\mathcal{D}W_i) \operatorname{vec}(\mathcal{D}W_j)^\top] = \mathbb{E}[a_{i-1} a_{j-1}^\top \otimes g_i g_j^\top]. \tag{14}$$

**Diagonal version.** We give an expression for the diagonal of $F_{i,i}(w)$ here. The diagonal of $F(w)$ follows immediately afterwards. Let $a_{i-1}^2$ and $g_i^2$ be the element-wise product of $a_{i-1}$ and $g_i$ respectively. Then, in vectorized form,

$$\text{diag}\,(F_{i,i}(w)) = \mathbb{E}[\text{vec}((a_{i-1}^2)(g_i^2)^\top)],$$

where $(a_{i-1}^2)(g_i^2)^\top$ is the outer product of $a_{i-1}^2$ and $g_i^2$.

### C.2 Convolutional networks

In order to write down the Fisher matrix for convolutional networks, it suffices to only consider convolution layers as the pooling and response normalization layers typically do not contain (many) trainable weights. We focus our analysis on a single layer. Much of the presentation here follows (Grosse & Martens, 2016; Luk & Grosse, 2018).

A convolution layer $l$ takes as input a layer of activations $a_{j,t}$ where $j \in \{1, \dots, J\}$ indexes the input map and $t \in \mathcal{T}$ indexes the spatial location. $\mathcal{T}$ here denotes the set of spatial locations, which we typically take to be a 2D-grid. We assume that the convolution here is performed with a stide of 1 and padding equal to the kernel radius $R$, so that the set of spatial locations is shared between the input and output feature maps. This layer is parameterized by a set of weights $w_{i,j,\delta}$, where $i \in \{1, \dots, I\}$ indexes the output map and $\delta \in \Delta$ indexes the spatial offset. The numbers of spatial locations and spatial offsets are denoted by $|\mathcal{T}|$ and $|\Delta|$ respectively. The computation of the convolution layer is given by

$$z_{i,t} = \sum_{\delta \in \Delta} w_{i,j,\delta} a_{j,t+\delta}. \tag{15}$$

The pre-activations $z_{i,t}$ are then passed through a nonlinear activation function $\phi_l$. The log-likelihood derivatives of the weights are computed through backpropagation:

$$\mathcal{D}w_{i,j,\delta} = \sum_{t \in \mathcal{T}} a_{j,t+\delta} \mathcal{D}z_{i,t}.$$

Then, the Fisher matrix here is

$$\mathbb{E}[\mathcal{D}w_{i,j,\delta} \mathcal{D}w_{i',j',\delta'}] = \mathbb{E}\left[\left(\sum_{t \in \mathcal{T}} a_{j,t+\delta} \mathcal{D}z_{i,t}\right)\left(\sum_{t' \in \mathcal{T}} a_{j',t'+\delta'} \mathcal{D}z_{i',t'}\right)\right].$$

**Diagonal version.** To give the diagonal version, it will be convenient for us to express the computation of the convolution layer in matrix notation. First, we represent the activations $a_{j,t}$ as a $J \times |\mathcal{T}|$ matrix $A_{l-1}$, the pre-activations $z_{i,t}$ as a $I \times |\mathcal{T}|$ matrix $Z_l$, and the weights $w_{i,j,\delta}$ as a $I \times J|\Delta|$ matrix $W_l$. Furthermore, by extracting the patches surrounding each spatial location $t \in \mathcal{T}$ and flattening these patches into column vectors, we can form a $J|\Delta| \times |\mathcal{T}|$ matrix $A_{l-1}^{\text{exp}}$ which we call the expanded activations. Then, the computation is Eqn. 15 can be reformulated as the matrix multiplication

$$Z_l = W_l A_{l-1}^{\text{exp}}.$$

Readers familiar with convolutional networks can immediately see that this is the Conv2D operation.

At a specific spatial location $t \in \mathcal{T}$, consider the $J|\Delta|$-dimensional column vectors of $A_{l-1}^{\text{exp}}$ and $I$-dimensional column vectors of $Z_l$. Denote these by $a_{l-1}^{(:,t)}$ and $z_l^{(t)}$ respectively. The matrix $W_l$ maps $a_{l-1}^{(:,t)}$ to $z_l^{(t)}$. In this case, we find ourselves in the exact same setting as the feed-forward case given earlier. The diagonal is simply

$$\mathbb{E}\left[\text{vec}\left((a_{l-1}^{(:,t)})^2 (\mathcal{D}z_l^{(t)})^2\right)\right]$$

## D   Kronecker-factored approximate curvature (K-FAC)

In Section 4, we compared the diagonal approximation of the Fisher matrix to the Kronecker-factored approximate curvature (K-FAC) (Martens & Grosse, 2015) approximation of the Fisher matrix. We give a brief overview of the K-FAC approximation in the case of feed-forward networks.

Recall that the Fisher matrix for a feed-forward network is a $L \times L$ matrix where each of the $(i, j)$ blocks are given by Eqn. 14. Consider the diagonal $(i, i)$ blocks. If we approximate the activations $a_{i-1}$ and pre-activation derivatives $g_i$ as statistically independent, we have

$$F_{i,i}(w) = \mathbb{E}[\text{vec}(\mathcal{D}W_i) \text{vec}(\mathcal{D}W_i)^\top] = \mathbb{E}[a_{i-1}a_{i-1}^\top \otimes g_i g_i^\top] \approx \mathbb{E}[a_{i-1}a_{i-1}^\top] \otimes \mathbb{E}[g_i g_i^\top].$$

Let $A_{i-1} = \mathbb{E}[a_{i-1}a_{i-1}^\top]$ and $G_i = \mathbb{E}[g_i g_i^\top]$. The K-FAC approximation $\hat{F}$ of the Fisher matrix $F$ is

$$\hat{F} = \begin{bmatrix} A_0 \otimes G_1 & & & 0 \\ & A_1 \otimes G_2 & & \\ & & \ddots & \\ 0 & & & A_{L-1} \otimes G_L \end{bmatrix}.$$

The K-FAC approximation of the Fisher matrix can be summarized in the following way: (1) keep only the diagonal blocks corresponding to individual layers, and (2) make the probabilistic modeling assumption where the activations and pre-activation derivatives are statistically independent.

# E    SUPPLEMENTARY EXPERIMENTAL RESULTS

In this section, we give additional validation accuracy results to complement Table 1. The additional regimes we experiment with are: BatchChange, Multiplicative, and K-FAC. For BatchChange, we use **SB** for 50 epochs and **LB** for 150 epochs. We perform this experiment to verify the argument given in Smith et al. (2017) that adapting from a small-batch regime to large-batch regime can close generalization gap. While Table 2 shows that BatchChange attains good validation accuracy numbers comparable to **SB**, this is not a preferred solution as it is not strictly LB training during the whole process which sacrifices the benefits of LB training in the early stages of training.

The other experiments we report here are Multiplicative and K-FAC. Multiplicative stands for multiplying gradients with Gaussian noise with constant diagonal convariance structure. This idea comes from (Hoffer et al., 2017). K-FAC means we use the K-FAC approximation instead of diagonal Fisher noise in Algorithm 1.

**Table 2:** Validation accuracy results on classification tasks using BatchChange, Multiplicative, K-FAC. For reader's convenience, we report again the result of Diag-F+GBN

| Dataset | Network | SB | BatchChange | Multiplicative | K-FAC | Diag-F+GBN |
|---------|---------|-----|-------------|----------------|-------|------------|
| CIFAR-10 | VGG16 | 93.25 | 93.18 | 90.98 | 93.06 | $93.15 \pm 0.05$ |
| CIFAR-100 | VGG16 | 72.83 | 72.44 | 68.77 | 71.86 | $71.94 \pm 0.14$ |
| CIFAR-10 | ResNet44 | 93.42 | 93.02 | 91.28 | 92.81 | $92.72 \pm 0.15$ |
| CIFAR-100 | ResNet44x2 | 75.55 | 75.16 | 71.98 | 73.84 | $74.10 \pm 0.18$ |

We give the training plots over epochs, the experimental results with the square root learning scaling scheme in the following table and plots,

**Table 3:** Validation accuracy results on classification tasks where Diag F+GBN uses square-root scaling learning rate.

| Dataset | Network | SB | LB+GBN | Diag-F+GBN |
|---------|---------|-----|--------|------------|
| CIFAR-10 | VGG16 | 93.25 | 91.6 | 92.9 |
| CIFAR-100 | VGG16 | 72.83 | 69.1 | 71.5 |
| CIFAR-10 | ResNet44 | 93.42 | 91.7 | 92.6 |
| CIFAR-100 | ResNet44x2 | 75.55 | 72.8 | 73.6 |

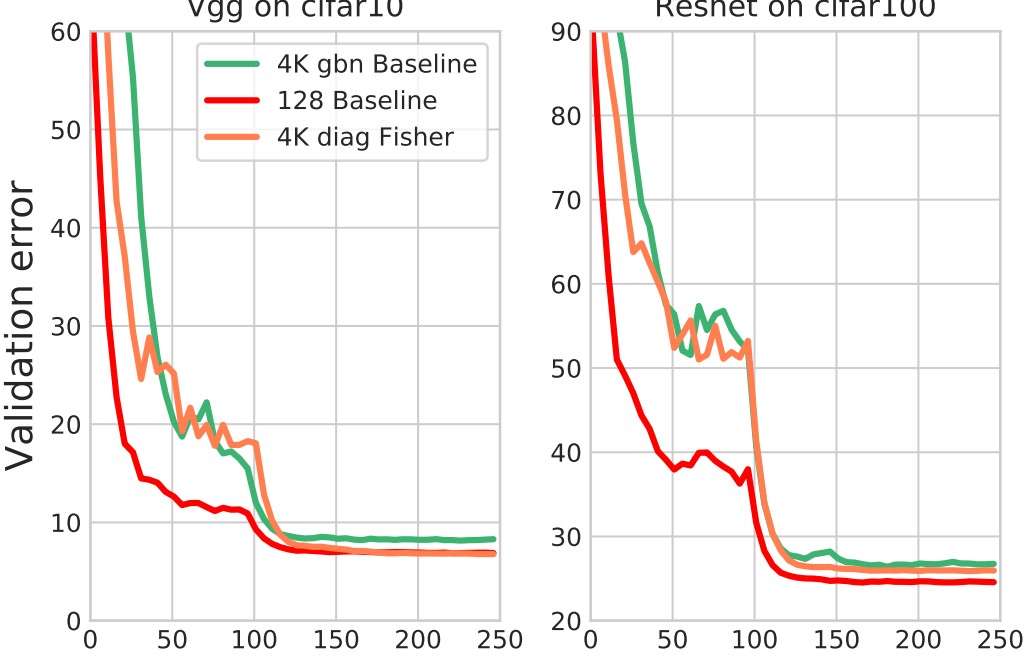

**Figure 4:** Validation error over number of epochs. **Left**: Training on CIFAR-10 with VGG-16. **Right**: Training on CIFAR-100 with resnet44. The reason why 4K with diagonal Fisher converges faster than the 4K GBN baseline in the beginning is because we need to use a larger learning rate for 4K GBN to get the best generalization performance, which makes it actually converge a bit slower in the beginning. If they share the same learning rate scheme, 4K GBN would converge a bit faster because it is less noisy.

