# OpenReview forum: "Exploring Curvature Noise in Large-Batch Stochastic Optimization"
_ICLR.cc/2019/Conference_

### Official Review · AnonReviewer2 · 2018-10-31
**Interesting and insightful theory but weak experimental setup**

**Rating:** 5
**Confidence:** 5

**Review:**

It has previously been observed that training deep networks using large batch-sizes leads to a larger generalization gap compared to the gap when training with a relatively small batch-size. This paper proposes to add noise sampled from diagonal "empirical" Fisher matrix to the large batch gradient as a method for closing the generalization gap. The authors motivate the use of empirical Fisher for sampling noise by arguing that the covariance of gradients from small batch-sizes can be seen as approximately equal to a scaled version of the Fisher matrix. It is then pointed out that using the Fisher matrix directly to sample noise could in principle close the generalization gap but would lead to slow converegence similar to SGD with a small batch-size. The authors then claim that the convergence speed is better when noise is sampled from the diagonal Fisher matrix instead of the full Fisher matrix. This claim is proven in theory for a convex quadratic loss surface and experiments are conducted to empirically verify this claim both in the quadratic setting are for realistic deep networks. Finally an efficient method for sampling noise from the diagonal empirical Fisher matrix is proposed.

Comments:
I think the paper is very well written and the results are presented clearly. In terms of novelty, I found the argument about convergence using diagonal Fisher being faster compared with full Fisher quite interesting, and its application for large batch training to be insightful.

As a minor comment, for motivating theorem 3.1, it is pointed out by the authors that the diagonal Fisher acts as an approximation of the full Fisher and hence their regularization effects should be similar while convergence should be faster for diagonal Fisher. As a caveat, I think the authors should also point out that the convergence rate would be best when C is set to 0 in the result of the theorem. This implies no noise is used during SGD updates. However, this would imply the regularization effect from the noise will also vanish which would lead to poor generalization.


However, there is a crucial detail that makes the main argument of the paper weak. In the main experiments in section 4.3, for the proposed large batch training method, the authors mention that they use a small batch-size of 128 for the first 50 epochs similar to Smith et al (2017) and then switch to the large batch-size of 4096, at which point, the learning rate is linearly scaled proportional to the large batch-size with a warmup scheme similar to Goyal et al (2017) and ghost batch normalization is used similar to Hoffer et al (2017). The former two tricks have individually been shown on their own to close the generalization gap for large batch-size training on large datasets like ImageNet. This paper combines these tricks and adds noise sampled from the diagonal Fisher matrix on top when switching to large batch-size after epoch 50 and reports experiments on smaller datasets-- MNIST, Fashion MNIST and the CIFAR datasets. Finally, the accuracy numbers for the proposed method is only marginally better than the baseline where isotropic noise is added to the large batch-size gradient. For these reasons, I do not consider the proposed method a significant improvement over existing techniques for closing the generalization gap for large batch training.

There is also a statement in the paper that is problematic but can be fixed by re-writing. In the paper, empirical Fisher matrix, as termed by the authors in the paper, refers to the Fisher matrix where the target values in the dataset is used as the output of the model rather than sampling it from the model itself as done for computing the true Fisher matrix. This empirical (diagonal) Fisher matrix is used to sample noise which is added to the large batch gradient in the proposed method. It is mentioned that the covariance of the noise in small batch SGD is exactly same as the empirical Fisher matrix. This claim is premised on the argument that the expected gradient (over dataset) is unconditionally roughly 0, i.e., throughout the training. This is absolutely false. If this was the case, gradient descent (using full dataset) should not be able to find minima and this is far from the truth. Even if we compare the scale of expected gradient to the mini-batch gradient (for small batch-size), the scale of these two gradients at any point during training (using say small batch-size SGD) is of the same order. I am saying the latter statement from my personal experience. The authors can verify this as well.

Overall, while I found the theoretical argument of the paper to be mostly interesting, I was dissapointed by the experimental details as they make the gains from the proposed method questionable when considered in isolation from the existing methods that close the generalization gap for large batch training.

---

> ### Author Response · Authors · 2018-11-06
> **Clarifying experimental setup in Sec. 4.3**
>
> We apologize for the confusion in writing in Section 4.3 and the issues that stemmed from it. We clarify our experimental setup here. All of the results concerning LB=4096, the last three columns presented in Table 1 are strictly LB during the entire training process. We did not use SB=128 for the first 50 epochs of training.
>
> We did perform an initial experiment where we used SB for first 50 epochs and LB for 150 epochs to suggest that the noise is relevant in the early stages of training. The results of this particular experiment are reported as “BatchChange” in Appendix F. We iterate that this is not a preferred solution; it is not strictly LB training during the entire process and hence it sacrifices the benefits of LB training in the early stages of training.
>
> In upcoming versions of this paper, we will modify Section 4.3 along with other sections of the paper to improve clarity and presentation of the paper. We will address your other comments and concerns in later rebuttals.

---

> > ### Comment · AnonReviewer2 · 2018-11-08
> > **Re: Clarifying experimental setup in Sec. 4.3**
> >
> > I see. But there were still two other tricks used-- the linear scaling scheme for learning rate and ghost batch-normalization, and the former has been shown to close the generalization gap on its own (Goyal et al 2017). Could you also report the improvement using diagonal Fisher conditioning but without the linear learning rate scaling scheme (may be on CIFAR-100)? I think it should not take long to run. According to the claim, the Fisher noise that is being added to the large batch gradient should lead to similar generalization as the base case, while improving convergence as pointed out by theory. So I am curious about where most of the gain is coming from-- the linear learning rate scaling scheme or the proposed Fisher noise.

---

> > > ### Author Response · Authors · 2018-11-08
> > > **About the linear scaling scheme and gbn**
> > >
> > > In the table1 where we reported for the baseline (LB+GBN), we also used the linear scaling scheme. It shows that the linear scaling scheme can't close the generalization gap on its own, at least without other tricks used in (Goyal et al 2017). The only difference between the baseline in our work and the (LB+GBN) in (Hoffer et al 2017) is the learning rate scheme (linear scaling vs. square root scaling). Hence, the gain from the linear learning rate scaling can be observed from the improvement of our numbers and theirs. But we still think it is interesting to see how diagonal Fisher works without the linear learning rate scaling scheme. We can report it in the upcoming revision.
> > >
> > > As for the GBN, it was shown in the (Hoffer et al 2017) that it needs more iterations to close the generalization gap on its own, which is against the motivation of LB training. We reported the numbers of (LB+gbn+noise) because it produces the best results without training longer but we think the gain from curvature noise is still clear and more importantly consistent with the convergence analysis.
> > >
> > > Another detail is the Isotropic noise in table1 is scaled by the Fisher trace norm so it is still curvature noise because it depends on the network parameters.

---

> > > > ### Comment · AnonReviewer2 · 2018-11-08
> > > > **Re: About the linear scaling scheme and gbn**
> > > >
> > > > I agree about GBN and the extended training regime that it is against the goal of training faster using large batch size. But at least it does show from a research point of view that the generalization gap can be closed when using large batch-size. However I find the baseline reported in Hoffer et al to be slightly weak compared to what I have seen in my experience. So I believe the gap is still present even when using their tricks and the problem is completely solved yet.
> > > >
> > > > I will wait for your experiments using diagonal Fisher without linear learning rate scheme.

---

> > > > > ### Author Response · Authors · 2018-12-04
> > > > > **Results without linear scaling learning rate.**
> > > > >
> > > > > We added the result with square root learning rate in the Appendix E. We also verbose the results here.
> > > > >
> > > > > Dataset    |  Network   |    SB     |  LB+GBN | Diag-F+GBN |
> > > > > Cifar10     |    VGG16    |  93.25   |    91.6      |  **92.9**     |
> > > > > Cifar100   |    VGG16    |  72.83   |    69.1      |  **71.5**     |
> > > > > Cifar10     |  Resnet44  |  93.42   |    91.7      |  **92.6**     |
> > > > > Cifar100   |Resnet44x2|  75.55   |    72.8      |  **73.6**     |

---

> ### Author Response · Authors · 2018-11-27
> **Response to Reviewer2**
>
> We would like to thank the reviewer for their feedback and suggestions to improve this paper. We address the reviewer’s concerns:
>
> -“As a caveat, I think the authors should also point out that the convergence rate would be best when C is set to 0 in the result of the theorem. This implies no noise is used during SGD updates. However, this would imply the regularization effect from the noise will also vanish which would lead to poor generalization.”
>
> The reviewer completely understands our point here. We do thank the reviewer for raising this point and in the latest version of our paper, we added this remark right after Theorem 3.1.
>
> -”the learning rate is linearly scaled proportional to the large batch-size with a warmup scheme similar to Goyal et al (2017) and ghost batch normalization is used similar to Hoffer et al (2017). The former two tricks have individually been shown on their own to close the generalization gap for large batch-size training on large datasets like ImageNet.”
>
> The learning rate scheduling in Goyal et al, 2017 (linear scaling w.r.t. batch-size) and Hoffer et al, 2017 (square-root scaling w.r.t. batch-size) indeeds helps generalization in their experiments. However, the recent paper of [1] shows that there is no optimal learning-rate scaling scheme. As long as the learning rate scheduling is consistent for all LB methods, we can accurately measure the gain in performance from utilizing diagonal Fisher. In the latest version of our paper, we added square-root scaling in Table 3 of Appendix E for completeness purposes.
>
> -”Finally, the accuracy numbers for the proposed method is only marginally better than the baseline where isotropic noise is added to the large batch-size gradient.”
>
> We thank the reviewer for bring into attention this confusion in writing. In Table 1, Isotropic + GBN is not exactly the baseline with isotropic Gaussian noise. In the main body of the text, what isotropic means is isotropic Gaussian noise scaled by trace of Fisher matrix. We apologize for this lack of clarity and have revised the writing in the most recent version. What this means is that in Eqn. 2 of the paper, we choose the covariance matrix D to be square-root of trace of Fisher and not the identity matrix. In fact, this is an important point in the paper: the choice of a covariance matrix D in Eqn. 2 has strong implications for both optimization and generalization performance. The reported numbers in Table 1 demonstrates that if we choose D to be Tr(F), we improve over the baseline (taken from Hoffer et al, 2017) but not as good as our proposed method in the last column.
>
> We add that if we choose the covariance matrix D to be identity (isotropic Gaussian noise), then the performance is significantly worse.
>
> -“Concerning expected gradient over joint distribution on dataset is approximately zero”
>
> We thank the reviewer for raising this point and we change the writing accordingly to remove previous confusions.
>
> Again, we appreciate the insightful comments made by this reviewer. In light of the changes that we made suggested by the reviewer as well as the contributions in our meta-review, we would appreciate if the reviewer can increase their score.
>
> [1]: Shallue, Christopher J., et al. "Measuring the Effects of Data Parallelism on Neural Network Training." arXiv preprint arXiv:1811.03600 (2018)

---

### Official Review · AnonReviewer1 · 2018-11-02
**Algorithm derivation is reasonable and interesting**

**Rating:** 6
**Confidence:** 4

**Review:**

Summary:
This paper proposes the method which improves the generalization performance of large-batch SGD by adding the diagonal Fisher matrix noise.
In the theoretical analysis, it is shown that gradient descent with the diagonal noise is faster than it with the full-matrix noise on positive-quadratic problems.
Moreover, the effectiveness of the method is verified in several experiments.

Comments:
The idea of the proposed method is based on the following observations and assumptions:

- Stochastic gradient methods with small-batch can be regarded as a gradient method with Fisher matrix noise.
- The generalization ability is comparable between diagonal Fisher and full Fisher matrix.
- Gradient method with diagonal Fisher is faster than that with full Fisher matrix.
This conjecture is theoretically validated for the case of quadratic problems.

In short, the algorithm derivation seems to be reasonable and the derived algorithm is executable.
Moreover, experiments are well conducted and the results are also good.


Minor comment:
- There is a typo in the next line of Eq. (2):
\nabla_{M_L} (\theta_k)} -> \nabla_{M_L} L(\theta_k)}

In addition, the notation "l_i" is not defined at this time.

---

> ### Author Response · Authors · 2018-11-27
> **Response to Reviewer1**
>
> We would like to thank the reviewer for the positive comments regarding our work.
>
> “There is a typo in the next line of Eq. (2): \nabla_{M_L} (\theta_k)} -> \nabla_{M_L} L(\theta_k)}”
> We have corrected this typo in the latest version of our paper
>
> Since the reviewer’s positive assessment of our paper and in consideration of the novel contributions of this paper given in our meta-review, we would very much appreciate if the reviewer can increase their score.

---

### Official Review · AnonReviewer3 · 2018-11-02
**curvature noise for large batch training in DNNs**

**Rating:** 5
**Confidence:** 4

**Review:**

In this paper, the authors propose a method to close the generalization gap that arises in training DNNs with large batch. The author reasons about the effectiveness in SGD small batch training by looking at the curvature structure of the noise. Instead of using the naïve empirical fisher matrix, the authors propose to use diagonal fisher noise for large batch SGD training for DNNs. The proposed method is shown empirically to achieve both comparable generalization and the training speedup compared to small batch training. A convergence analysis is provided for the proposed method under convex quadratic setting.

The idea of exploring the curvature information in the noise in SGD has been studied in (Hoffer et al. 2017). The difference between this approach and the proposed method in the paper is the use of diagonal fisher instead of the empirical fisher. Although there is convergence analysis provided under convex quadratic setting, I feel that the motivation behind using diagonal fisher for faster convergence is not clear to me, although in the experiment part, the comparison of some of the statistics of diagonal fisher appear similar to the small batch SGD. The intuition of using diagonal fisher for faster convergence in generalization performance is still missing from my perspective.

In the convergence analysis, as there is a difference between the full fisher and diagonal fisher in the Tr(C’AC) term. It would be interesting to see the effect of how this term play on convergence rate, and also how this term scale with batch size. But this is more of a minor issue as we are mostly caring about its generalization performance which is different from optimization error convergence.

In the experiments section, the authors claim that noise structure is only important for the first 50 epochs. But it would be better if the authors could show experimental results of using the same training method all the way during the experiment. The experiments are conducted on MNIST and CIFAR10 and 100, which I feel is a bit insufficient for a paper that deals with generalization gap in large batch. As in large batch training, we care more about bigger dataset such as ImageNet, and hence I would expect results reported on various models on ImageNet. Another interesting thing to show would be the generalization error over epochs for different methods, which could give a more detailed characterization of the behavior of different methods.

Overall, I feel the motivation and intuition behind the proposed method is not clear enough and experimental studies are not sufficient for understanding the behavior of the proposed method as an empirical paper.

---

> ### Author Response · Authors · 2018-11-27
> **Response to Reviewer3**
>
> We like to thank the reviewer for the comments and suggestions to improve this paper. We address the reviewer’s concerns:
>
> -“The idea of exploring the curvature information in the noise in SGD has been studied in (Hoffer et al. 2017). The difference between this approach and the proposed method in the paper is the use of diagonal fisher instead of the empirical fisher.”
>
> Hoffer et al, 2017 discussed intrinsic curvature noise in SGD (as do many other recent papers such as Smith et al, 2017, Chaudhari et al, 2017, etc.). The proposed solutions in all these papers did not explicitly implement any form of empirical Fisher gradient noise. The main approach implemented in Hoffer et al, 2017 is the use of Ghost-Batch Normalization (GBN). GBN, similar to usual BN, should be thought as an architectural modification rather than incorporating curvature noise information. In addition, the training procedure is elongated for LB. However, extending the training regime for LB is against the very goal of using LB in the first place. In contrast, we show that using diagonal Fisher noise preserves the desirable convergence performance of LB training per parameter update and significantly improves generalization performance of LB without training longer.
>
> - “Although there is convergence analysis provided under convex quadratic setting, I feel that the motivation behind using diagonal fisher for faster convergence is not clear to me, although in the experiment part, the comparison of some of the statistics of diagonal fisher appear similar to the small batch SGD. The intuition of using diagonal fisher for faster convergence in generalization performance is still missing from my perspective.”
>
> The intuition can be understood in the following way. In Fig 1, we notice that the empirical full Fisher update is orthogonal to the loss curvature. Thus, adding full Fisher noise to the gradients gives large perturbation in the high curvature direction, which leads to a higher expected training loss. In comparison, only taking the diagonal results in a smaller perturbation in the high curvature direction, which leads to a smaller expected training loss.
>
> This is quantified in Theorem 3.1 for the convex quadratic setting. The overall convergence rate of the bound is O(1/k) but the constant is Tr(C^TAC). The difference between using full Fisher (C=\sqrt{A}) and diagonal Fisher is (C=\sqrt{diag A}) is exactly the difference between their Frobenius norms. We show that this carries over to the deep learning setting: in Figure 3a), we show that the Frobenius norm of full Fisher is much larger than that of diagonal Fisher. Finally, in Fig 3c), we showed that FB (full-batch) + diagonal Fisher attains much faster training than FB + full Fisher, which verifies the above-mentioned statement.
>
> -“In the convergence analysis, as there is a difference between the full fisher and diagonal fisher in the Tr(C’AC) term. It would be interesting to see the effect of how this term play on convergence rate, and also how this term scale with batch size. But this is more of a minor issue as we are mostly caring about its generalization performance which is different from optimization error convergence.”
>
> The difference between the Frobenius norm of the Fisher matrix and diagonal Fisher matrix is independent of the batch-sizes involved. That being said, the batch-sizes are used in the coefficient \sqrt{N-M}{NM} before the diagonal Fisher term in Algorithm 1.
>
> -“The experiments are conducted on MNIST and CIFAR-10/100, which I feel is a bit insufficient for a paper that deals with generalization gap in large batch. As in large batch training, we care more about bigger dataset such as ImageNet, and hence I would expect results reported on various models on ImageNet.“
>
> The reason that all of our experiments were conducted on smaller models and datasets such as MNIST, CIFAR-10/100 are due to constraints in computing resources. We did not have the computing power or budget to run experiments on ImageNet. However, we feel that the empirical analysis given in our paper addresses key research questions concerning both convergence and generalization of LB training.
>
> -“Another interesting thing to show would be the generalization error over epochs for different methods, which could give a more detailed characterization of the behavior of different methods.”
> Please see Appendix E in the latest version of our paper.
>
> In light of the changes that we made suggested by the reviewer as well as the contributions in our meta-review, we would appreciate if the reviewer can reconsider their score.

---

> > ### Comment · AnonReviewer3 · 2018-11-28
> > **Re: Response to Reviewer3**
> >
> > Thanks for the reply. I still have some concerns after reading the response.
> > 1. I could see the motivation of using diagonal Fisher noise from the optimization perspective, however it remains unclear to me how it helps for generalization than Fisher noise. And we know in deep learning, optimization does not guarantee generalization in general. The convergence theory part mostly only cares about optimization, so I would be more interested in seeing the motivation from the generalization perspective for diagonal Fisher noise.
> > 2. Same with Reviewer 2, I would hope to see experiments without linear scaling. As the proposed Fisher noise is to account for the noise component, it is expected to have similar behavior to small-batch SGD.
> > 3. Also, I feel that ImageNet is a standard dataset for studying generalization of large-batch training as also reported in the recent literature (Hoffer et. al 17, Goyal et. al 17). Therefore, it is of much interest as the paper is proposing an empirical approach for closing the generalization gap.

---

> > > ### Author Response · Authors · 2018-11-30
> > > **Further response to Reviewer 3**
> > >
> > > We thank the reviewer for pointing out more potential concerns. We address them as follows:
> > >
> > > - Motivation for using diagonal Fisher to improve generalization
> > >
> > > In the convex-quadratic setting, we provided a theoretical analysis in Appendix B of how choosing different noise covariance structures in Eqn. 2 impacts the generalization error. The choice of noise covariance equal to Fisher and equal to identity are extreme cases and diagonal Fisher represents a “middle ground”. The Hellinger distance is much smaller when diagonal Fisher is chosen compared to identity leading to a tighter generalization bound in Eqn. 11.
> > >
> > > Moreover, the experiments show that taking the diagonal preserves some generalization benefits of full Fisher. An example of this is reported in Table 1 where we trained ResNet-44 model on CIFAR-10, the validation accuracy for full Fisher was 93.22 while diagonal Fisher was 92.72 (whereas the baseline LB + GBN is 91.92).
> > >
> > > - Results without linear scaling
> > >
> > > We have included extra experiments with square-root scaling in Appendix E. Indeed, in the experiments, full Fisher has the same behavior as SB and no LR scaling is required for it. However, for diagonal Fisher, since the optimization property is closer to LB, the learning rate needs to be scaled (either linear or square root) to get optimal performance.
> > >
> > > - ImageNet experiments
> > >
> > > Well-designed experiments on MNIST and CIFAR-10/100 have a good track record of generalizing to large datasets such as ImageNet.  In the recent systematic studies of LB training in [1], many of the experimental results are qualitatively consistent between different datasets (ranging from MNIST to Open Images) and architectures (ranging from Fully Connected to ResNet-50) within epoch-budget regimes.
> > >
> > > We want to iterate that the central focus of our paper is not an empirical study on closing generalization gap of LB. Rather, we discovered adding diagonal Fisher gradient noise to LB gives significantly better optimization performance than full Fisher (and also SB) while they roughly share the same gradient variance and improving generalization of LB. We gave a full justification in the convex-quadratic setting.
> > >
> > > We thank the reviewer again for responding in such a timely manner. Please let us know if there are more concerns with any other parts of the paper.
> > >
> > > [1]: Shallue, Christopher J., et al. "Measuring the Effects of Data Parallelism on Neural Network Training." arXiv preprint arXiv:1811.03600 (2018)

---

> > > > ### Comment · AnonReviewer3 · 2018-12-04
> > > > **Re: Further response to Reviewer 3**
> > > >
> > > > The generalization analysis in Appendix B gives a generalization bound using uniform stability, however it remains to be explained why using diagonal fisher from the generalization perspective (which is also mentioned in the paper).
> > > > And for learning rate scaling, I do not really get why the optimization performance serve as a rule for choosing the stepsize for good generalization.
> > > > Experiment wise, as the theory part does not explicitly justify the use of diagonal fisher information for generalization, experiments on standard ImageNet would be necessary to demonstrate its superior performance.

---

> > > > > ### Author Response · Authors · 2018-12-05
> > > > > **Further response to Reviewer 3**
> > > > >
> > > > > We thank the reviewer for the timely response.
> > > > >
> > > > > Regarding using diagonal Fisher for generalization, we first explain the stability argument given in Appendix B in greater detail. The generalization bound is given by Eqn. 11 where the right-hand side is given by a Hellinger distance between two probability distributions. Since we are working over the convex-quadratic setting, we have Ornstein-Uhlenbeck processes and the Hellinger distance is given by Eqn. 13. As t is going to infinity, the term of importance are the determinants of the diffusion matrices $\Sigma_S(t)$. In page 14 of our paper, we gave the formula for the determinant of $\Sigma_S(t)$ for the cases where the noise is either identity or full Fisher. For full Fisher, we see that there is less dependence on the data (more robust) and for identity, we have more dependence on the data (given by the denominator term) and so we choose diagonal Fisher as a middle ground. That explains why diagonal Fisher gives a better generalization than LB.
> > > > >
> > > > > Furthermore, from a heuristic standpoint, we provided experiments on the maximum eigenvalue of the Hessian and marginal variance of the gradients. In Figure 3b), we find that the marginal variance of the gradients for LB + diag F is much larger than LB and similar to SB/LB + full Fisher. In Figure 3c), we find that the maximum eigenvalue of LB + diag F is smaller than the baseline LB + GBN. While the connection is not completely explicit, the fact that such heuristics are correlated with generalization performance has been discussed extensively in the literature; for example in [1, 2, 3, 4].
> > > > >
> > > > > From [5], the time-to-target-validation (generalization) is explained purely by optimization considerations. So optimization performance is definitely an important sign of step-size selection.
> > > > >
> > > > > [1] Chaudhari, Pratik, et al. "Entropy-sgd: Biasing gradient descent into wide valleys." arXiv preprint arXiv:1611.01838(2016)
> > > > > [2] Chaudhari, Pratik, and Stefano Soatto. "Stochastic gradient descent performs variational inference, converges to limit cycles for deep networks." 2018 Information Theory and Applications Workshop (ITA). IEEE, 2018.
> > > > > [3] Keskar, Nitish Shirish, et al. "On large-batch training for deep learning: Generalization gap and sharp minima." arXiv preprint arXiv:1609.04836 (2016).
> > > > > [4]:  Sagun, Levent, et al. "Empirical Analysis of the Hessian of Over-Parametrized Neural Networks." arXiv preprint arXiv:1706.04454 (2017).
> > > > > [5]: Shallue, Christopher J., et al. "Measuring the Effects of Data Parallelism on Neural Network Training." arXiv preprint arXiv:1811.03600 (2018)

---

> > > > > > ### Comment · AnonReviewer3 · 2018-12-05
> > > > > > **Re: Further response to Reviewer 3**
> > > > > >
> > > > > > Thanks for the reply.
> > > > > >
> > > > > > The generalization theory analysis part does not serve as a complete justification to me but it is more of a motivation of using fisher curvature noise. Therefore, I would still be expecting more comprehensive experiments
> > > > > >
> > > > > > I have looked at the paper[5] mentioned by the author, however I do not seem to find the argument of explaining generalization by optimization. It seems that only optimization budget is considered in that paper.

---

> > > > > > > ### Author Response · Authors · 2018-12-05
> > > > > > > **Further response to Reviewer 3**
> > > > > > >
> > > > > > > We thank the reviewer for the quick response. Regarding the generalization analysis, we agree that it is not a complete picture. However, within the current uniform stability framework that we work with in Appendix B, there is little or nothing more we can do. It is not possible to explicitly measure the difference of the determinant of the diffusion matrix $\Sigma_S(t)$ between taking full Fisher and diagonal Fisher. In that case, we would like to ask what type of theoretical analysis the reviewer wishes to see? It would be certainly interesting to provide rigorous guarantees for LB + diagonal Fisher and LB + full Fisher in the non-convex deep learning setting. However, we believe that this would be far too ambitious a task and the state of current deep learning theory offers no tools for us to do so.  Additionally, the focus of the paper is the optimization difference among choices of curvature noise.
> > > > > > >
> > > > > > > In [1], all plots are present as out-of-sample-error (generalization) vs. steps. We can see that the model degradation of LB can be explained by not having enough optimization updates, which suggests a better optimization scheme is at least an important sign of better generalization.
> > > > > > >
> > > > > > > [1]: Shallue, Christopher J., et al. "Measuring the Effects of Data Parallelism on Neural Network Training." arXiv preprint arXiv:1811.03600 (2018)

---

> > > > > > > > ### Comment · AnonReviewer3 · 2018-12-05
> > > > > > > > **Re: Further response to Reviewer 3**
> > > > > > > >
> > > > > > > > Thanks for the reply.
> > > > > > > > I agree with the author that currently generalization theory is missing and therefore that's the reason why I would expect more comprehensive empirical study of the proposed method.
> > > > > > > > As for experiments in [1], more optimization step updates imply better generalization, which is also observed in other literature such as [2]. However, I don't see the reasoning for how step size could be chosen according to the optimization behavior of the algorithm. And may I know what does it mean by "better optimization scheme"? If it means optimization convergence, I do not see how a faster optimization convergence leads to a better generalization for deep learning.
> > > > > > > >
> > > > > > > > [2] Hoffer, Elad, Itay Hubara, and Daniel Soudry. "Train longer, generalize better: closing the generalization gap in large batch training of neural networks." Advances in Neural Information Processing Systems. 2017.

---

> > > > > > > > > ### Author Response · Authors · 2018-12-06
> > > > > > > > > **Further response to Reviewer 3**
> > > > > > > > >
> > > > > > > > > Thanks for the response.
> > > > > > > > >
> > > > > > > > > I agree that more comprehensive experiments needed if the paper just focuses on the large batch generalization aspect. However, the main contribution of the paper is the surprising role of variance we mentioned in the summary of contribution. Generalization improvement is a bonus of the curvature noise where we gave a detailed discussion. We will revise the paper in the camera-ready version to make these points more clear in the writing. Additionally, running batch size 4K with ResNet44 or VGG16 on ImageNet is really out of our capability (In [1], to fit batch size 8192 with ResNet50, they use 256 GPUs and 50Gbit of network bandwidth).
> > > > > > > > >
> > > > > > > > > Better optimization scheme could either mean a lower train loss or faster convergence. Slower convergence in the epoch budget (our setting) could lead to a higher train loss. So it should be considered as a criterion of step-size selection.
> > > > > > > > >
> > > > > > > > > [1]: Priya Goyal, et al. "Accurate, Large Minibatch SGD: Training ImageNet in 1 Hour." arXiv preprint arXiv:1706.02677 (2017)

---

> > > > > > > > > > ### Comment · AnonReviewer3 · 2018-12-11
> > > > > > > > > > **Re: Further response to Reviewer 3**
> > > > > > > > > >
> > > > > > > > > > Under large-batch sgd, as shown in Figure 1 in Hoffer et. al's paper [1], although different batch size could achieve similar training loss, it could be observed that the generalization gap still exists. Therefore, I still have doubt about how the optimization convergence could be used as criterion for step-size selection. According to the author, SGD with different batch sizes would be selected to have the similar step-size since their training loss under same epochs look similar.
> > > > > > > > > >
> > > > > > > > > >
> > > > > > > > > >
> > > > > > > > > > [1] Hoffer, Elad, Itay Hubara, and Daniel Soudry. "Train longer, generalize better: closing the generalization gap in large batch training of neural networks." Advances in Neural Information Processing Systems. 2017.

---

> > > > > > > > > > > ### Author Response · Authors · 2018-12-11
> > > > > > > > > > > **Further response to Reviewer 3**
> > > > > > > > > > >
> > > > > > > > > > > We agree that the optimization can't be used alone to choose the step-size. All we want to say is optimization is one of the criterions to choose the step size especially in the case that the difference in the training loss can be observed.
> > > > > > > > > > >
> > > > > > > > > > > But how to choose the best step-size is orthogonal to our paper. Our main contribution is to point out that not only the variance but also the covariance structure matters in the optimization convergence by giving the following example: the gradients of LB + diag Fisher, LB + full Fisher and SB have roughly the same variance (see Fig 3b). However, in Fig 3c, the convergence of LB + diagonal Fisher is much faster than LB + full Fisher. In fact, it is roughly equal to the convergence of LB which has much smaller variance. We performed theoretical analysis in the convex case, conducted careful experiments and found the LB + diagonal Fisher also gives a better generalization performance.

---

> ### Comment · Area_Chair1 · 2018-12-11
> **ImageNet**
>
> I would just like to comment on the ImageNet requirement: the resources required to perform extensive experiments on ImageNet are out of reach for many authors and as such cannot be made a requirement for a submission. Although I agree it would be nice to get results on larger datasets, it is unfair to penalize a work for not including them.

---

### Author Response · Authors · 2018-11-27
**Summary of contributions and changes.**

Summary of central contributions:

-The role of variance: Variance-reduction techniques are commonly used in machine learning to improve convergence. However, convergence is not only related to the variance but also the covariance structure. In this paper, we found some surprising counter-intuitive phenomenon. In particular, the following three methods: the gradients of LB + diag Fisher, LB + full Fisher and SB have roughly the same variance (see Fig 3b). However, in Fig 3c, the convergence of LB + diagonal Fisher is much faster than LB + full Fisher. In fact, it is roughly equal to the convergence of LB. Moreover, over the convex-quadratic setting, we give a theoretical convergence guarantee for FB (full-batch) + diagonal Fisher and FB + full Fisher in terms of their Frobenius norms. We demonstrate that the Frobenius norm analysis carries over to deep neural networks in our experiments.

-In a recent comprehensive study on LB training [1], the authors showed that epoch-budget training favors SB regime and is disadvantageous for LB. Moreover, the work of [2] shows that LB’s generalization performance can be remedied by extending the number of epochs trained. In this paper, we demonstrated that using diagonal Fisher noise allows us to improve LB generalization performance within an epoch-budget training (and in addition, retaining LB’s fast convergence per iteration).

-Analyzing optimization and generalization together: In this paper, we examined both the optimization and generalization performance of LB training in unison rather than separately. Recent works (for example, [3]) indicates that optimization and generalization in deep learning cannot be decoupled.

[1]: Shallue, Christopher J., et al. "Measuring the Effects of Data Parallelism on Neural Network Training." arXiv preprint arXiv:1811.03600 (2018)
[2]: Hoffer, Elad, Itay Hubara, and Daniel Soudry. "Train longer, generalize better: closing the generalization gap in large batch training of neural networks." Advances in Neural Information Processing Systems. 2017.
[3]: Neyshabur, Behnam, et al. "Exploring generalization in deep learning." Advances in Neural Information Processing Systems. 2017.

============================================================================================

Summary of changes:

-Revised Section 3.3
-Minor rewriting of Section 4.3 as suggested by Reviewer 2 to remove previous confusions about experimental setup
-Added paragraph at end of Theorem 3.1 discussing how taking C=0 impacts optimization and generalization. This was suggested by Reviewer 2
-Added Table 3 in the Appendix using square-root scaling scheme. This was requested by Reviewer 2
-Added Figure 4 in Appendix E showing generalization performance of different regimes with respect to number of epochs. This was requested by Reviewer 3

---

### Meta-Review · Area_Chair1 · 2018-12-14
**Issues with the experiments and case study not corresponding to the actual algorithm**

**Confidence:** 4
**Recommendation:** Reject

**Metareview:**

Dear authors,

Your proposition of adding a noise scaling with the diagonal of the gradient covariance to the updates as a middle-ground between the identity and the full covariance is interesting and tackles the timely question of the links between optimization and generalization.

However, the reviewers had concerns about the experiments that did not reveal to which extent each trick had an influence.
I would like to add that, even though the term Fisher is used for both the true Fisher and tne empirical one, these two matrices encore very different kind of information. In particular, the latter is only defined when there is a dataset. Hence, your case study  (section 3.2) which uses the true Fisher does not apply to the empirical Fisher.

I encourage the authors to pursue in this direction but to update the experimental section in order to highlight the impact of each technique used.